# Study on prediction of blasting cracking radius of liquid $CO_2$ in coal

**Jinzhang Jia**[1,2]**, Yumo Wu**[1,2]*, **Dan Zhao**[3]**, Bin Li**[4]**, Dongming Wang**[1,2]

**1** College of Safety Science and Engineering, Liaoning Technical University, Fuxin Liaoning, China, **2** Key Laboratory of Thermal Dynamic Disaster Prevention and Control of Ministry of Education, Liaoning Technical University, Huludao Liaoning, China, **3** Faculty of Civil Engineering and Architecture, Zhanjiang University of Science and Technology, Zhanjiang, Guangdong, China, **4** School of Environmental and Chemical Engineering, Shenyang Ligong University, Shenyang, China

* 13614067811@163.com

**Data Availability Statement:** All relevant data are within the paper and its Supporting Information files.

**Funding:** This work was partly supported by the National Natural Science Foundation of China (grant number 52174183), the Natural Science

## Abstract

In this study, we sought to improve the efficiency of coal seam gas extraction, master the characteristics of different factors on the liquid carbon dioxide ($CO_2$) phase change blasting cracking radius, and effectively predict the hole spacing. In this study, we used ANSYS/LS-DYNA numerical simulation software to predict the crack radius of liquid $CO_2$ phase change blasting combined with orthogonal design scheme. The results showed that the primary and secondary factors affecting the fracture radius of liquid $CO_2$ phase change blasting were in ground stress, gas pressure, coal firmness coefficient, and gas content. The fracture radius decreased with the increase of in ground stress and decreased with the increase of gas pressure, coal firmness coefficient, and gas content, which was linear. A prediction model for predicting the cracking radius of liquid $CO_2$ phase change blasting based on four groups of different factors was established. Through the double verification of numerical simulation and field industrial test, the cracking radius of liquid $CO_2$ phase change blasting ranged from 2 m to 2.5 m. The maximum error of numerical simulation was 2.8%, and the maximum error of field industrial test was 5.93%.

## 1. Introduction

Coal is an indispensable fuel energy for many countries and will continue to play its role of "ballast" in maintaining the smooth operation of the economy for a long time to come. In the new era of energy systems, the scientific use of coal is key to achieving carbon neutrality [1, 2]. Coal mine gas disaster accidents are still occurring [3, 4], however, and are one of the main factors affecting the safe mining of coal seams. Gas extraction is the most commonly used technical means to prevent gas disasters. Therefore, improvements in the efficiency of gas extraction in high gas and low permeability coal seams are urgently needed [5, 6]. The methods to improve the efficiency of coal seam gas drainage mainly include coal seam water injection [7], hydraulic cracking [8], hydraulic slotting [9], hydraulic cavitation [10], hydraulic punching [11], high-pressure water jet [12], and deep hole pre-splitting blasting [13]. These methods have been applied in engineering practice. Although they can improve the efficiency of gas drainage, they also have some shortcomings. Hydraulic measures have problems such as large

Foundation of Liaoning Province (grant number 2019-MS-162) and the Scientific Research Project of Guangdong Provincial Department of Education —Young Innovative Talents Project (grant number 2022KQNCX141). The first and second funders are first author Jinzhang Jia and the third funder is third author Dan Zhao. There was no additional external funding received for this study.

**Competing interests:** The authors have declared that no competing interests exist.

water consumption and pore blocking [14], whereas deep hole blasting has problems such as high pollution and high risk [15]. Therefore, many scholars have proposed the use of liquid $CO_2$ phase change blasting antireflection technology. This method has low investment and high efficiency, and has been gradually promoted and applied in the field of coalbed methane development [16, 17].

Liquid $CO_2$ phase change blasting antireflection technology is one of the safest and most reliable technical means to improve the efficiency of gas extraction. Its characteristics fully meet the development needs of various countries on the coal industry. Because of the complexity of coal seam structure, many factors have led to the blasting cracking radius, which directly affect gas extraction [18, 19]. Therefore, this topic has always been a focus of industry researchers to identify the factors affecting the liquid $CO_2$ phase change blasting cracking and to master methods to improve prediction accuracy. Zhou et al. [20] considered the effect of in ground stress on the radius of liquid $CO_2$ phase change cracking, and obtained the single-hole blasting radius under the influence of in ground stress by numerical simulation. Zhao Baoyou et al. [21] compared the numerical simulation with the experimental results, studied the influence of gas pressure and coal firmness coefficient on the blasting effect, and found that two groups of single factors had a positive effect on the fracture radius. Yan et al. [22, 23] conducted the numerical simulation of $CO_2$ cracking based on the extended finite element method and the TNT equivalent method, and developed a fluid-solid coupling model. Zhang et al. [24] studied the liquid $CO_2$ phase change blasting experiment and found that the crack propagation process was affected by both $CO_2$ volume and surrounding rock stress field. Wang et al. [25] theoretically analyzed the phase change cracking process of liquid $CO_2$ and established a mathematical model of coal seam cracking pressure and cracking influence range. Then they conducted a numerical simulation of $CO_2$ phase change cracking. The results showed that the influence radius of cracking was 2.55–2.70 m. Jia et al. [26], based on the gray correlation analysis method, the effects of different factors on the phase change cracking effect of liquid $CO_2$ were studied. The results showed that the phase change cracking effect of liquid $CO_2$ was positively correlated with the changes of gas pressure, elastic modulus, fracture hole diameter, and detonation peak pressure, and this effect was negatively correlated with the changes of in ground stress and compressive strength, and was almost independent of tensile strength. Nilson [27] established the integral equation of rock mass crack propagation under the action of fluid-gas interaction and visually analyzed the quasi-static effect of explosive gas and the relationship between coal crack propagation. Soliman [28] and EI-Rabaa [29] have shown that the tensile strength theory can effectively predict the fracture development. Draou et al. [30] proposed two methods: The first is to observe the change of vertical stress, and according to the compaction principle, porosity changes exponentially with vertical stress. The second is the change of porosity rate caused by the change of vertical stress, which plays a guiding role in predicting formation gap and fracture pressure. To more accurately calculate the energy used for phase change expansion cracking of liquid $CO_2$, Dong Qingxiang et al. [31] introduced the energy utilization rate using the TNT equivalent method, which provided theoretical support for the establishment of a numerical model of gas explosion cracking. Through this analysis, at present, many scholars have conducted some research on liquid $CO_2$ phase change cracking technology, but most of the existing research has been limited to the influence of single factor on the cracking radius, and the calculation methods are different. Because of the complex geographical environment of the underground, many factors affect the cracking radius of liquid $CO_2$ phase change, and the propagation law of cracks after blasting is relatively complex. This requires the application of reasonable methods to conduct multifactor coupling analysis, and the calculation method of liquid $CO_2$ phase change cracking radius is optimized to enhance the prediction accuracy and improve the controllability of crack propagation after blasting.

In light of this research, under the condition of considering the coupling of four groups of influencing factors, such as ground stress, gas pressure, coal body firmness coefficient, and gas content, we used ANSYS-LSDYNA numerical simulation software to analyze the primary and secondary order and influence degree of the factors affecting the cracking radius of liquid $CO_2$ phase change blasting by combining range and variance analysis. We used an orthogonal design method to optimize the numerical simulation scheme and reduce the simulation workload. We processed the simulation data using multiple regression analysis and established a prediction model of the cracking radius of liquid $CO_2$ phase change blasting under the influence of various factors. This research method provides a theoretical reference for the prediction of cracking radius of different blasting methods.

## 2. Theory and method

### 2.1 Basic principle of liquid $CO_2$ phase change blasting

The characteristics of $CO_2$ at ambient temperature and pressure are easy to liquefy and not to burn. The critical condition of $CO_2$ is 7.38 MPa and the temperature is 31.1˚C. If the temperature exceeds the critical temperature, the liquid $CO_2$ absorbs heat and enters the supercritical state [32]. Supercritical fluid has the advantages of a gas-liquid two-phase fluid at the same time. It has super flow, transmission, and permeability. It can maximize the conductivity of natural fractures and improve the conductivity of fractures. It has unique advantages in fracturing. The basic principle of liquid $CO_2$ cracking technology is as follows: liquid $CO_2$ is loaded into the cracking device and placed in the drilled hole, and the equipped cracking device is connected with the mine initiator to form a closed loop. After detonation, the hot cartridge will release heat energy instantaneously under the excitation of a current. This increase in heat makes the liquid $CO_2$ enter the supercritical state immediately. In this state, the rapid gasification volume of liquid $CO_2$ increases by more than 700 times in an instant, and the pressure in the cracking device increases rapidly. When the gas pressure in the cracking device reaches a predetermined value, the preset shear disc at the venting head breaks, and the gaseous $CO_2$ erupts into the coal around the cracking device through the guide hole. This instantly produces strong impact pressure. The cracking device is shown in Fig 1.

The damage process of shock wave, stress wave, and detonation gas on coal rock mass caused by liquid $CO_2$ phase change blasting is complicated [33]. First, the gas produced by the explosion will instantaneously act on the wall of the blasting hole, and at the same time, it will create a strong impact energy on the wall of the coal rock mass, thus forming a large area of crushing zone. The explosion shock wave following the explosion is hindered by the rock mass in the crushing area, and it gradually decays into a stress wave. The conduction of this stress wave will produce the tangential stress acting on the rock mass, which is greater than the tensile strength of the original rock mass. Thus, the original rock mass forms a new expansion crack and forms the blasting middle zone. The detonation gas produced in the later stage also develops gradually, so that the radial cracks continues to expand. The area formed by the intersection of radial cracks and circumferential cracks is called the crack propagation area. When the stress wave attenuates to the point at which it can no longer cause damage to the coal body, this part of the area that can propagate only in the form of seismic waves is called the vibration zone [34, 35], as shown in Fig 2. The propagation process of stress in coal can be described by Hooke 's law, as follows:

$$\rho \frac{\partial^2 u}{\partial t^2} = \rho \frac{\partial \sigma}{\partial x} + \frac{\partial \tau_{xy}}{\partial y} + \frac{\partial \tau_{xz}}{\partial z}, \tag{1}$$

$$\rho \frac{\partial^2 v}{\partial t^2} = \rho \frac{\partial \tau_{yx}}{\partial x} + \frac{\partial \sigma_y}{\partial y} + \frac{\partial \tau_{yz}}{\partial z}, \tag{2}$$

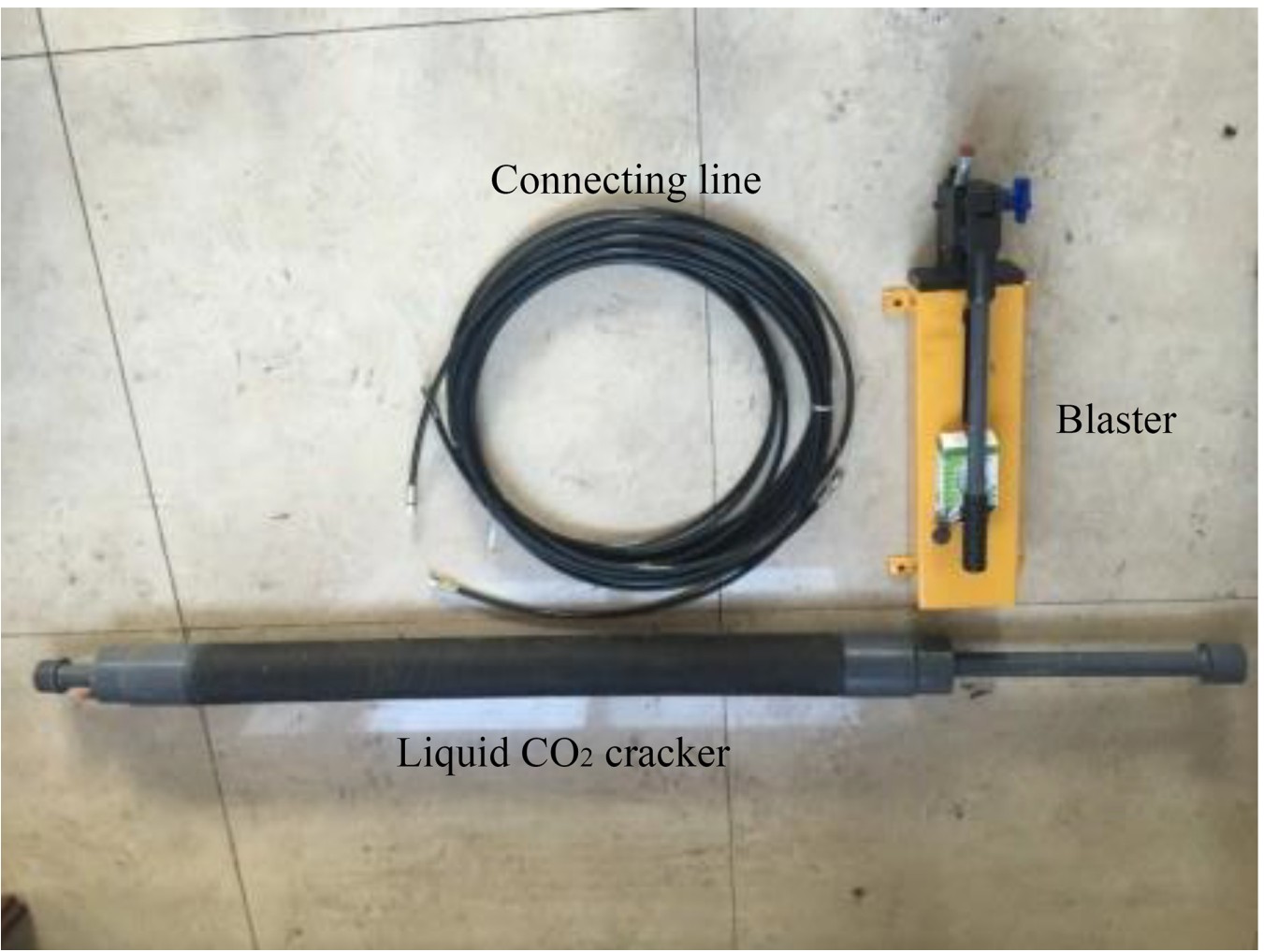

**Fig 1. Liquid CO₂ cracking device.**

$$\rho \frac{\partial^2 w}{\partial t^2} = \rho \frac{\partial \tau_{zx}}{\partial x} + \frac{\partial \tau_{zy}}{\partial y} + \frac{\partial \sigma_z}{\partial z}, \tag{3}$$

where $\rho$ is the density of coal, kg·m$^{-3}$; $u$, $v$, and $w$ are three displacement components of coal mass point, m; $\rho \frac{\partial^2 u}{\partial t^2}$, $\rho \frac{\partial^2 v}{\partial t^2}$, and $\rho \frac{\partial^2 w}{\partial t^2}$ are the three acceleration components of particle, m·s$^{-2}$; and $\sigma_x$, $\sigma_y$, $\sigma_z$, $\tau_{xy}$, $\tau_{xz}$, and $\tau_{yz}$ are six stress components of coal (rock) mass point.

In coal medium, it is generally believed that the propagation direction of plane stress wave is parallel to the x-axis. Thus, we have the following:

$$\left. \begin{array}{l} u = u(x, t), v = w = 0 \\ \varepsilon_x \neq 0, \varepsilon_y = \varepsilon_z = 0, \theta = \varepsilon_x \\ \sigma_x \neq 0, \sigma_y = \sigma_z \neq 0 \end{array} \right\}. \tag{4}$$

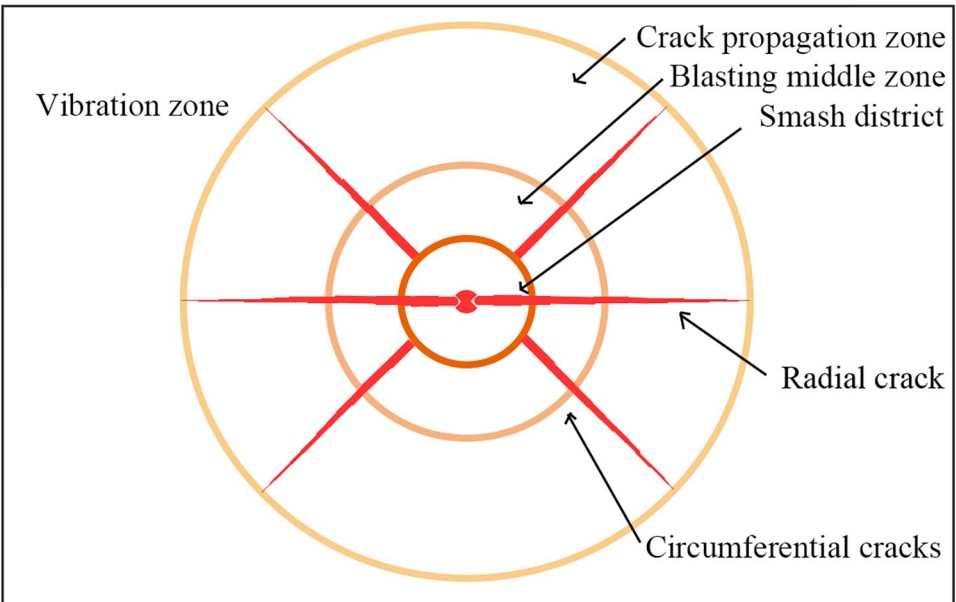

**Fig 2. Liquid CO$_2$ phase transition blasting cracking principle diagram.**

Eq (4) can be transformed as follows:

$$\frac{\partial^2 u}{\partial t^2} = c_P^2 \frac{\partial^2 u}{\partial x^2},$$

(5)

where $c_P^2 = \frac{(\lambda + 2G)}{\rho}$.

By Hooke 's law and plane wave, we have the following:

$$\left.\begin{array}{l} \sigma_x = (2G + \lambda)\dfrac{\partial u}{\partial x} \\[2mm] \sigma_x = \sigma_y = \lambda\dfrac{\partial u}{\partial x} \\[2mm] \tau_{xy} = \tau_{yz} = \tau_{zx} = 0 \end{array}\right\}.$$

(6)

The propagation process of the particle is mainly affected by longitudinal wave cp and transverse wave cs. The propagation velocity can be obtained by substituting the initial and boundary conditions into Eq (6):

$$c_p = \sqrt{\frac{1 + 2G}{\rho}} = \sqrt{\frac{E(1 - \mu)}{\rho(1 + \mu)(1 - 2\mu)}},$$

(7)

$$c_s = \sqrt{\frac{G}{\rho}} = \sqrt{\frac{E}{2\rho(1 + \mu)}}.$$

(8)

With the extension of action time, the stress wave energy produced by liquid CO$_2$ phase change blasting gradually weakens, and its effect on coal mass in the later stage of conduction becomes smaller and smaller. The attenuation law of stress peak in homogeneous coal

conforms to Eq (9) [36], as follows:

$$\sigma_r = \sigma_0 \left(\frac{r}{r_0}\right)^{-\alpha}.$$ 

(9)

## 2.2 TNT equivalent calculation of liquid CO$_2$ phase change blasting

In nature, liquid CO$_2$ phase transition blasting is categorized as a physical explosion, which is a physical change from the liquid phase to the gas phase in a short period of time. Its principle is not exactly the same as that of high-temperature and high-pressure gas produced at the moment of a TNT explosion, which is several times the volume of explosive, but the general process of the two is basically the same [37]. Therefore, to facilitate numerical simulation, TNT equivalent calculation method can be used to convert the burst energy of liquid CO$_2$ phase change cracking. The TNT equivalent calculation equation follows:

$$E_g = \frac{P_1 V}{K - 1}\left[1 - \left(\frac{P_2^{(K-1)/K}}{P_1}\right)\right] \times 10^3,$$ 

(10)

where $E_g$ is gas explosion energy, kJ; $P_1$ is the gas pressure in the blasting cracker, 275 MPa; $P_2$ is the standard atmospheric pressure, taken as 0.10108 MPa; $V$ is cracking volume, m$^3$; and $K$ is the adiabatic index of the medium, taking 1.295.

The approximate TNT equivalent $W$ of the energy released by the phase change explosion of liquid CO$_2$ is calculated as follows:

$$W = \frac{E_g}{Q},$$ 

(11)

where $Q$ is the explosion energy of 1 kg TNT explosive, taking 4250 kJ·kg$^{-1}$.

According to the calculation of TNT equivalent, the blasting energy per 1 kg of liquid CO$_2$ phase change blasting is the same as that of 397 g TNT explosive. In this study, the liquid volume of liquid CO$_2$ cracking was 1.48 kg, and the explosive equivalent was 588 g TNT explosive.

## 2.3 Orthogonal design of numerical simulation scheme

The orthogonal design method is to use the principle of orthogonality and mathematical statistics, from a large number of simulation data selected as representative data to simplify the simulation workload. The orthogonal table is used to divide the multiple factors of the test at different levels, and the number of simulations is reduced to obtain the optimal simulation results and achieve the most scientific simulation results [38]. Because of the variability of the external factors of coal, and different internal factors have different effects on the deformation and failure of coal. Therefore, in this study, we selected four main factors that had a significant influence on the cracking effect of liquid CO$_2$ phase change blasting, including ground stress, gas pressure, coal firmness coefficient (f), and gas content. Each factor was set to four levels, and an empty column was left for error analysis in practical application. We applied an orthogonal design scheme L16 (4$^5$), and realized 16 kinds of simulation comparison schemes. The specific settings are shown in Table 1.

## 2.4 Data processing

To analyze the primary and secondary order and influence degree of the factors affecting the cracking radius of liquid CO$_2$ phase change blasting, we used the data to calculate the mean point and then determined the range analysis. Through the range comparison, the primary

**Table 1. Simulation scheme orthogonal table.**

| Group | Ground stress (MPa) | Gas pressure (MPa) | Coal firmness coefficient | gas content ($m^3 \cdot t^{-1}$) | Blank column |
|---|---|---|---|---|---|
| a | 6 | 0.2 | 0.5 | 5 | 0 |
| b | 6 | 0.3 | 0.6 | 6 | 0 |
| c | 6 | 0.4 | 0.7 | 7 | 0 |
| d | 6 | 0.5 | 0.8 | 8 | 0 |
| e | 8 | 0.2 | 0.6 | 7 | 0 |
| f | 8 | 0.3 | 0.5 | 8 | 0 |
| g | 8 | 0.4 | 0.8 | 5 | 0 |
| h | 8 | 0.5 | 0.7 | 6 | 0 |
| i | 10 | 0.2 | 0.7 | 8 | 0 |
| j | 10 | 0.3 | 0.8 | 7 | 0 |
| k | 10 | 0.4 | 0.5 | 6 | 0 |
| l | 10 | 0.5 | 0.6 | 5 | 0 |
| m | 12 | 0.2 | 0.8 | 6 | 0 |
| n | 12 | 0.3 | 0.7 | 5 | 0 |
| o | 12 | 0.4 | 0.6 | 8 | 0 |
| p | 12 | 0.5 | 0.5 | 7 | 0 |

and secondary order of the influencing factors can be arranged intuitively. To make up for the lack of accuracy of range analysis, we used variance analysis to compare different confidence levels and confirmed the results. This improved the accuracy of the test. To better describe the significance and influence effect of the four groups of factors, we combined the orthogonal design method and the multiple linear regression analysis method to obtain the regression relationship of the test index—that is, the model to predict the cracking radius of liquid $CO_2$ phase change blasting.

## 3. Numerical simulation of liquid $CO_2$ phase change cracking

### 3.1 Establishment of numerical model

In this article, we used ANSYS-LSDYNA numerical simulation software to simulate the phase change cracking of liquid $CO_2$. We set the single-hole and double-hole sizes of the liquid $CO_2$ phase change blasting model to 6 m × 6 m and 16 m × 12 m, respectively, and set the aperture of the cracking hole to 94 mm. The model consists of three parts: explosive, air, and coal rock. The fluid-solid coupling algorithm is adopted. The explosive and air use the Euler algorithm, the coal rock uses the Lagrange algorithm, the unit uses the multi-material ALE algorithm, and the continuous medium model uses the ALG algorithm. The mesh of single-hole blasting is divided into 369411 units, and the number of nodes is 742344. The mesh of double-hole blasting is divided into 859197 units, and the number of nodes is 1724102. A uniform load of 12 MPa is applied to the top of the model, which is equivalent to the weight of a 900-meter rock layer. The lateral pressure coefficient is 1.5; that is, the horizontal stress is 18 MPa. The boundary conditions are: the surface of the coal-rock model is constrained, the sides of model Z are unidirectionally constrained, the Z direction is constrained, and the model boundary is a non-reflective interface boundary. The material parameters of coal and rock are shown in Table 2.

**Table 2. Coal rock material parameters.**

| Density ($g \cdot cm^{-3}$) | Elastic modulus (GPa) | Poisson ratio | Tensile strength (MPa) | Compressive strength (MPa) | Cohesion (MPa) |
|---|---|---|---|---|---|
| 1.54 | 1.74 | 0.3 | 0.84 | 2.2 | 2.5 |

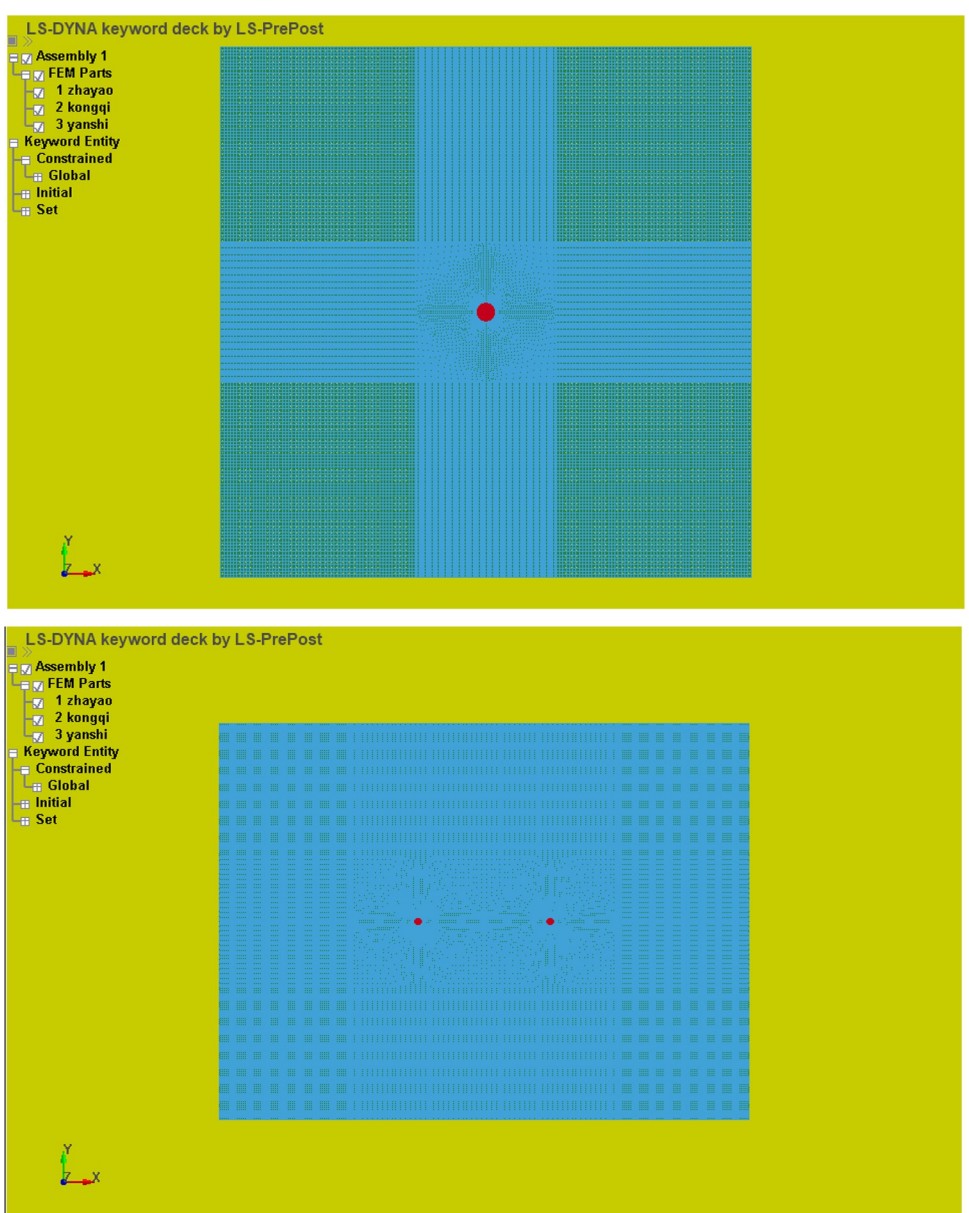

**Fig 3. Liquid CO$_2$ phase transition explosion model.** (a) Single hole, (b) Double hole.

We added *MAT_ADD_EROSION to control for the failure of the coal unit. When the unit exceeded the original load, it would fail and be deleted. Thus, we were able to intuitively express the development law of cracks after gas explosion. The meshing of single-hole and double-hole numerical models of liquid CO$_2$ phase change cracking is shown in Fig 3.

## 3.2 Numerical simulation results analysis

Because the stress and strain on coal under blasting is very complex, in order to show the propagation and distribution of the explosive stress field in the process of liquid CO$_2$ phase change blasting, the damage cloud map of the model at different times can be dynamically described by LS-PREPOST post-processing software. Through the concealment of explosives and air

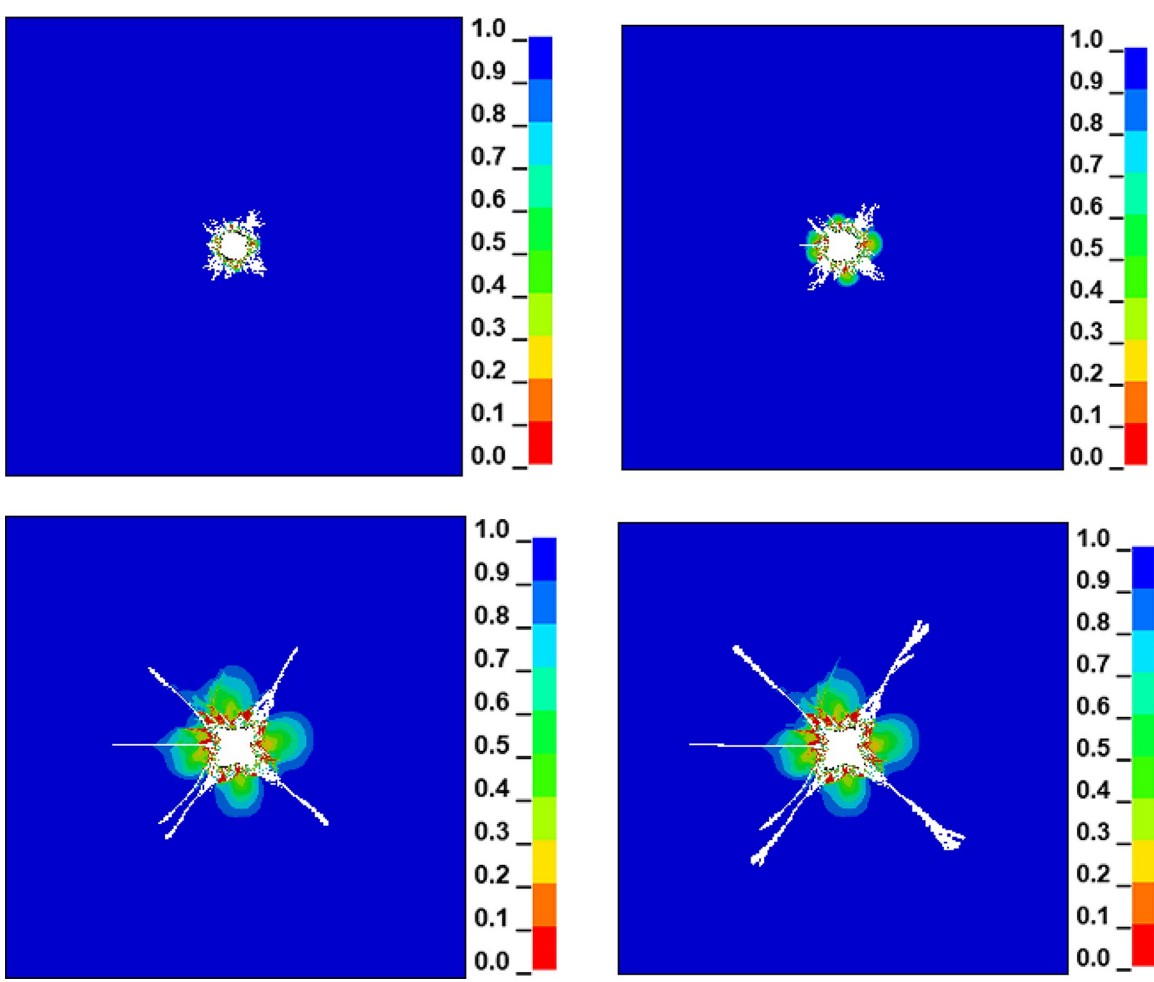

**Fig 4. Simulation evolution process of liquid $CO_2$ phase transition blasting.**

units, the effective stress of coal material can be understood more intuitively. Through the evolution simulation of the whole process of blasting, the coal damage distribution cloud map of the blasting effect at different times is obtained, as shown in Fig 4.

The fracture development law of liquid $CO_2$ phase change blasting in coal bodies is as follows: A strong stress wave will be generated in the borehole immediately after detonation in t = 0~20 us. Because the coal wall near the blasting hole absorbs a large amount of stress wave energy, the coal wall around the borehole is crushed over a large area; that is, a crushing area is formed. The main crack appears gradually when t = 20 us. When t = 35 us, branched cracks begin to appear, the area of the high pressure zone in the center of the borehole gradually increases, and new cracks gradually form. The stress wave continues to extend outward from the coal-rock crushing zone of the coal seam and gradually forms a coal-rock crack zone. When t = 20~80 us, the crack continues to expand. When t = 80 us, the damage zone of the fracture ring reaches its maximum, and the stress wave generated by the gas explosion has changed from a plastic to an elastic wave. The driving force of the crack to the distant extension is becoming smaller and smaller as high gas pressure is continuously attenuated. When t = 100us, the maximum axial stress at the tip of the extended crack is less than its own dynamic ultimate tensile strength, and the crack stops expanding. It can be seen that blasting gas damage to coal rock is a complex, dynamic evolution process.

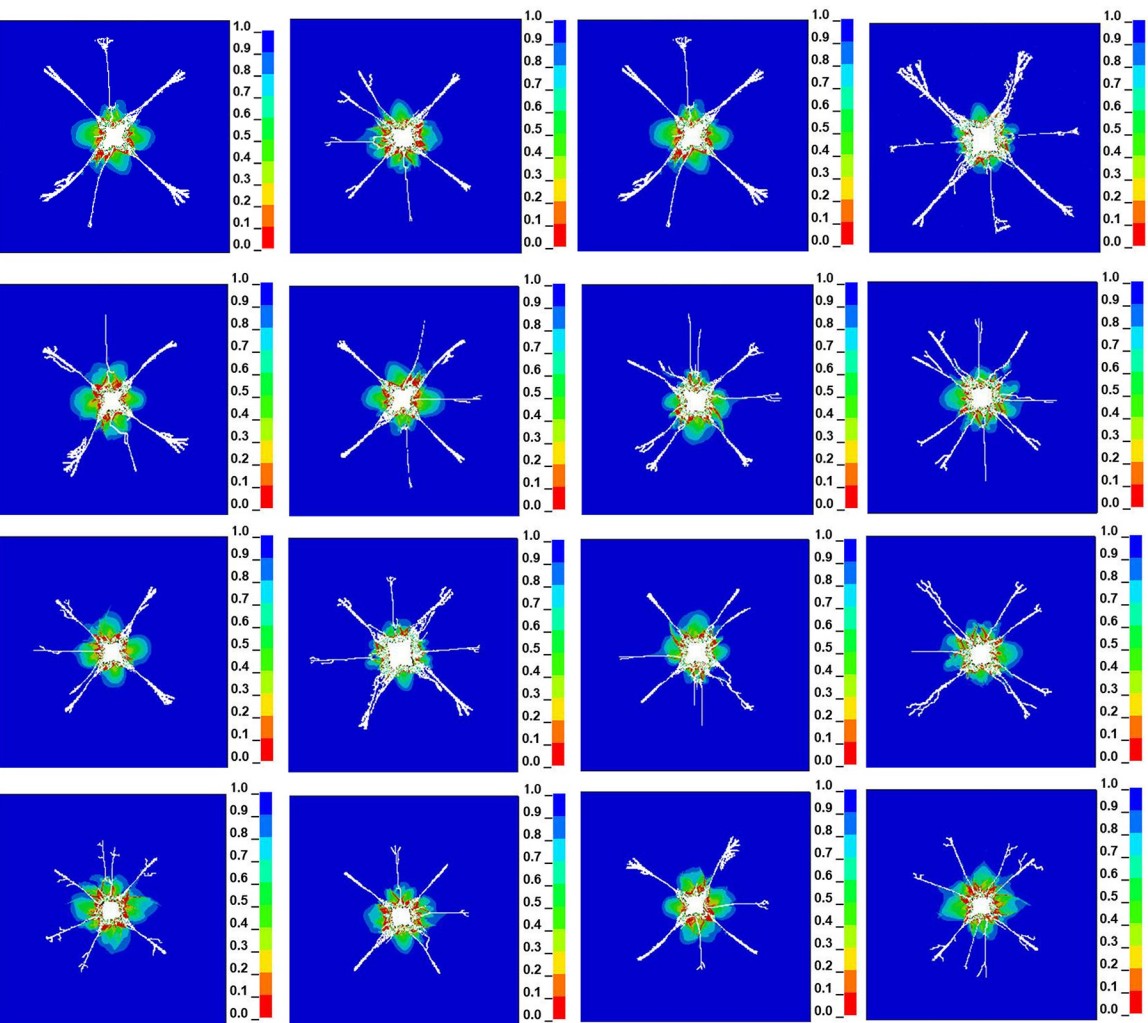

**Fig 5. Numerical simulation results of liquid $CO_2$ phase change blasting cracking.**

On the basis of the orthogonal design results, we conducted a numerical simulation of liquid $CO_2$ phase change blasting antireflection according to the 16 groups of representative simulation comparison schemes. The control equations and boundary conditions remained unchanged, and only the values of the four groups of factors in the orthogonal design changed. We used the LS-PrePost module for post-processing to observe the influence of different four groups of factors on the cracking radius (r). The specific numerical simulation results of liquid $CO_2$ phase change blasting antireflection are shown in Fig 5.

The numerical simulation results of liquid $CO_2$ phase change explosion antireflection are calculated by mean value and range analysis. The calculation results are shown in Table 3.

According to the range analysis in Table 3, the primary and secondary orders of factors affecting the cracking radius of liquid $CO_2$ phase change blasting are ground stress, gas pressure, coal firmness coefficient, and gas content. According to the mean point in Table 3, the data fitting was possible. The influence of various factors on the liquid $CO_2$ phase change blasting radius is shown in Fig 6.

According to an intuitive analysis, we found the following factors to have influenced the cracking radius of liquid $CO_2$ phase change blasting:

**Table 3. Range analysis of cracking radius.**

| Horizontal group | Ground stress (MPa) | Gas pressure (MPa) | Coal firmness coefficient | Gas content ($m^3 \cdot t^{-1}$) | Blank column |
|---|---|---|---|---|---|
| Mean point 1 | 2.325 | 2.000 | 2.025 | 2.050 | 2.125 |
| Mean point 2 | 2.175 | 2.050 | 2.100 | 2.075 | 2.075 |
| Mean point 3 | 2.050 | 2.150 | 2.125 | 2.125 | 2.100 |
| Mean point 4 | 1.875 | 2.225 | 2.175 | 2.175 | 2.125 |
| Range value | 0.450 | 0.225 | 0.150 | 0.125 | 0.050 |

1. The cracking radius of liquid $CO_2$ phase change blasting decreased with the increase of in ground stress. With the increase of in ground stress, the closure degree of original cracks in coal seam increased, which led to a decrease in the permeability of the coal seam and inhibited the development of cracks after phase change cracking.

2. The cracking radius of liquid $CO_2$ phase change blasting increased with the increase in gas pressure. The gas pressure in the coal body led to a decrease in the effective strength of the coal body, which was conducive to the development of tip cracks.

3. The cracking radius of liquid $CO_2$ phase change blasting increased with the increase in the coal firmness coefficient. The brittleness of coal with low hardness was large, and it was easy to break under the influence of a stress wave and formed a crushing circle when blasting occurred, which hindered the conduction of the stress wave and affected the further development of cracks.

4. The cracking radius of liquid $CO_2$ phase change blasting increased with the increase in gas content. The higher the gas content was, the more stable the stress gradient that caused the coal to break when the blasting occurred. As a result, the broken coal was continuously thrown out by the sufficient gas flow, and the cracks of the coal body continued to develop deeply.

We analyzed the results of orthogonal design of numerical simulation of liquid $CO_2$ phase change blasting by variance analysis, and the confidence levels were 90%, 95%, and 99%, respectively. The results are shown in Table 4.

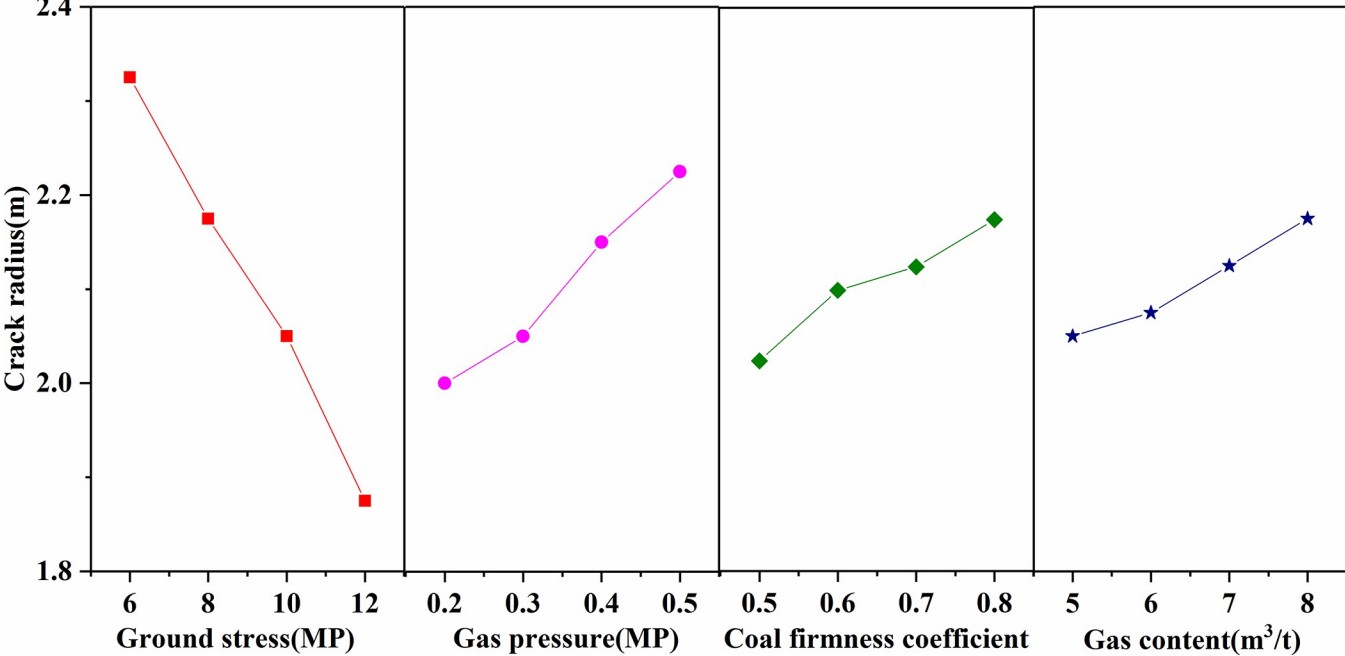

**Fig 6. Intuitive analysis of influence of factors on cracking radius.**

**Table 4. Variance analysis.**

| Influencing factor | | Ground stress (MPa) | Gas pressure (MPa) | Coal firmness coefficient | Gas content ($m^3 \cdot t^{-1}$) | Error |
|---|---|---|---|---|---|---|
| Square of deviance | | 0.437 | 0.122 | 0.047 | 0.037 | 0.01 |
| Degree of freedom | | 3 | 3 | 3 | 3 | 3 |
| F-ratio | | 62.429 | 17.429 | 6.714 | 5.286 | — |
| F marginal value | a = 0.1 | 5.390 | 5.390 | 5.390 | 5.390 | — |
| | a = 0.05 | 9.280 | 9.280 | 9.280 | 9.280 | — |
| | a = 0.01 | 29.50 | 29.50 | 29.50 | 29.50 | — |
| Significance | a = 0.1 | * | * | * | * | — |
| | a = 0.05 | * | * | * | — | — |
| | a = 0.01 | * | * | * | — | — |

From the F ratio in the orthogonal design variance analysis results of the liquid $CO_2$ phase change blasting antireflection numerical simulation, we concluded that the primary and secondary order of the influence intensity of each factor on the crack radius of liquid $CO_2$ phase change blasting was as follows: ground stress, gas pressure, coal body firmness coefficient, and gas content, which was consistent with the results of range analysis. For the cracking radius of liquid $CO_2$ phase change blasting, when the confidence of ground stress, gas pressure, and coal firmness coefficient was 90%, 95%, and 99%, the significance reached a high level, indicating that these three groups of factors had an obvious influence on the cracking radius. The significance of the gas content was shown only in the 90% confidence interval, which indicated that it had a lower impact on the fracture radius compared with the first three groups of factors. The sum of the squared deviations, however, was greater than the error group, which proved that the results of the orthogonal test were correct.

Through the analysis of each factor and the visual analysis of the influence of each factor on the crack radius of liquid $CO_2$ phase change blasting shown in Fig 6, it was evident that each influencing factor had a linear relationship with the simulation results. We set the ground stress as $x_1$, gas pressure as $x_2$, coal firmness coefficient as $x_3$, and gas content as $x_4$; the liquid $CO_2$ phase change blasting cracking radius was $y$. We obtained the data given in Table 5 by multiple regression analysis of the 16 groups of orthogonal test data given in Table 2.

The prediction model between y and x could be obtained by derivation, as shown in Eq (12):

$$y = 1.889 - 0.073x_1 + 0.791x_2 + 0.501x_3 + 0.042x_4, \tag{12}$$

where $R^2$ represents the goodness of fit, and the larger $R^2$, the better the fitting effect.

According to the data analysis in Table 5, the $R^2$ value is 0.982, indicating that the goodness of fit of the equation was very high. To further prove the rationality of the prediction model, we conducted a double-hole simulation comparison test and conducted the error analysis.

## 3.3 Double-hole verification of prediction model

To verify the rationality of the prediction model for predicting the fracture radius of liquid $CO_2$ phase change blasting, we selected two sets of measuring point data for the west return

**Table 5. Orthogonal design prediction model.**

| — | $b_0$ | $b_1$ | $b_2$ | $b_3$ | $b_4$ | $R^2$ |
|---|---|---|---|---|---|---|
| $y$ | 1.889 | -0.073 | 0.791 | 0.501 | 0.042 | 0.982 |

**Table 6. Prediction of hole spacing.**

| Group | Ground stress (MPa) | Gas pressure (MPa) | Coal firmness coefficient | Gas content ($m^3 \cdot t^{-1}$) | Pre—prediction of cracking radius (m) | Predict hole spacing (m) |
|---|---|---|---|---|---|---|
| 1 | 8.3 | 0.45 | 0.62 | 9.73 | 2.36 | 4.7 |
| 2 | 8.2 | 0.33 | 0.6 | 6.58 | 2.12 | 4.2 |

airway in the No. 5 mining area of Sihe Coal Mine. We substituted the data into the prediction model and obtained the predicted hole spacing of the two points, as shown in Table 6.

Using the predicted hole spacing obtained in Table 6, we established a double-hole model. The boundary conditions and parameter settings were consistent with the single-hole model. Only the size and hole spacing of the double-hole model changed. The hole spacing of a group of double-hole models was set to 4.7 m and 5.5 m; and two groups of two-hole model hole spacing models were set to 4.2 m, 5 m. The two-hole simulation results of liquid $CO_2$ phase change blasting cracking at the two points are shown in Fig 7.

From Fig 7A and 7C, it is evident that after the liquid $CO_2$ phase change blasting of two drilling holes with spacing of 4.7 m and 4.2 m, the crack propagation between the two holes formed a breakthrough near the center of the connection between the two holes. This cracking was mainly due to the superposition of the stress field generated by the blasting, which guided the initiation and development of the crack. As a result, the two holes cracked through and

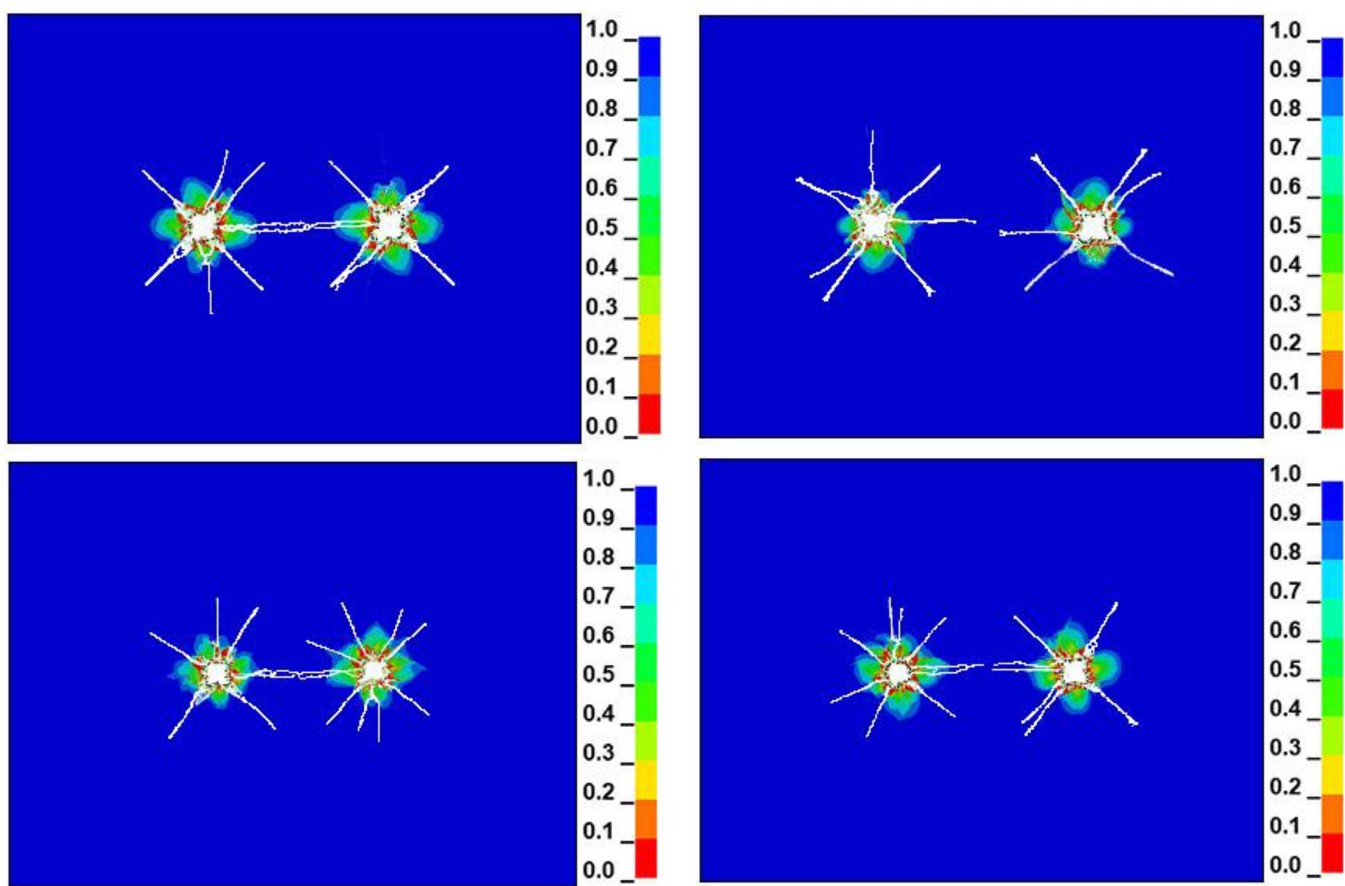

**Fig 7. Double-hole simulation diagram of liquid $CO_2$ phase change blasting.** (a) 4.7m, (b) 5.5m, (c) 4.2m, (d) 5.m.

formed a joint crushing zone. From Fig 7B and 7D, it is evident that after the two drilling holes with hole spacing of 5.5 m and 5 m were blasted by liquid $CO_2$ phase change, because of the large hole spacing, the superposition effect of the stress wave between the holes weakened. As a result, the cracks between the holes could not be connected, and finally, a relatively independent blasting fracture zone was formed. At the same time, the crack radius of the two measuring points of group 1 and group 2 could be measured to be 2.41 m and 2.18 m. After calculation, the simulation errors of the two measuring points were 2.1% and 2.8%, respectively. The simulation error was small, and the prediction model could be considered reasonable.

## 4. Field engineering application test

To further verify the accuracy of the numerical simulation of liquid $CO_2$ phase change blasting and the rationality of the prediction model, the industrial test of the liquid $CO_2$ phase change blasting was conducted in the west return airway of the No. 5 mining area of the Sihe Coal Mine, and the crack radius of the liquid $CO_2$ phase change blasting was measured according to the tracer gas $SF_6$ measurement method. The main instruments used for the tracer gas $SF_6$ determination are shown in Fig 8.

### 4.1 Drilling parameter design of test site layout

We arranged the two test groups at the test blasting site. In the first experimental group, drilling was conducted at 300 m away from the open-off cut in the west return airway, and the No. 1 cracking holes were arranged. Six tracer gas observation holes $A_1$–$A_6$ were set on both sides of the cracking holes. The second test group was arranged at 800 m away from the open-off cut in the west return airway, and the No. 2 cracking hole and six tracer gas observation holes $A_7$–$A_{12}$ were arranged. This arrangement of observation holes was consistent with that of the first test group, as shown in Fig 9.

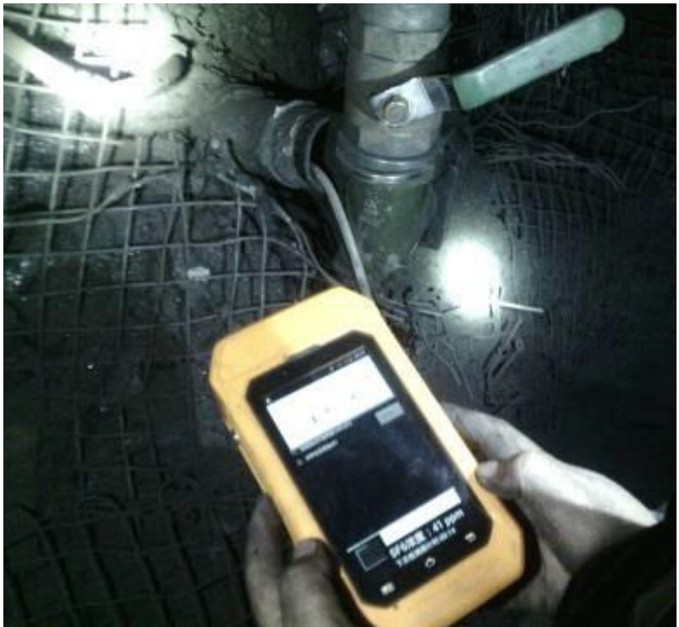
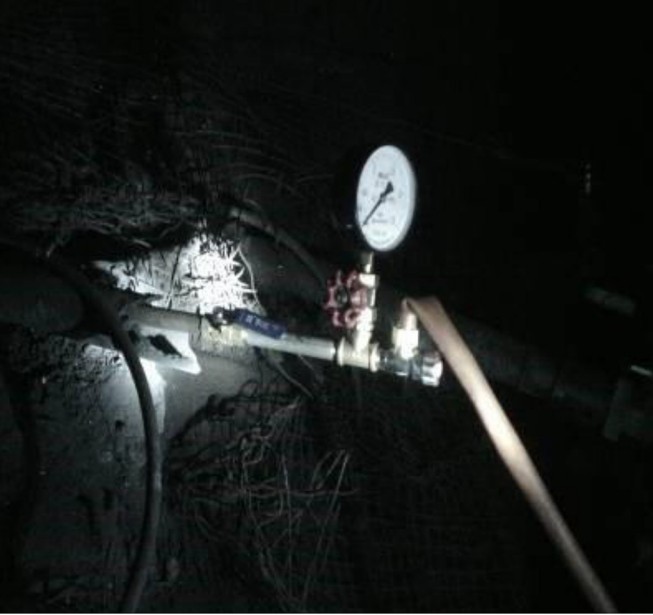

**Fig 8. Main instruments for SF₆ tracer gas determination.** (a) $SF_6$ gas concentration recorder, (b) $SF_6$ gas transmitters and digital pressure gauges.

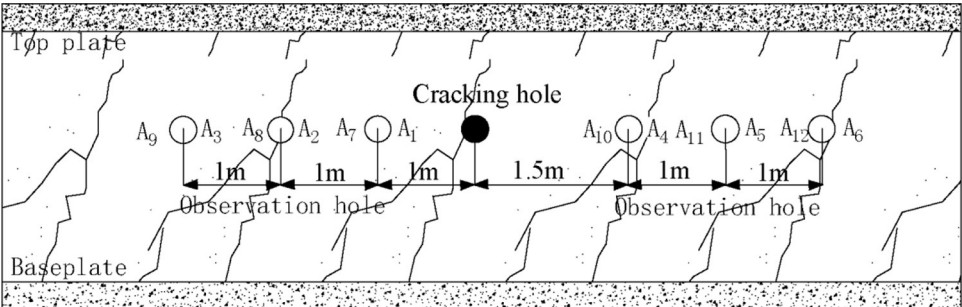

**Fig 9. Drilling layout in the test site.**

In the process of drilling construction, we first constructed 12 observation holes in the experimental group and then constructed the cracking holes. After the construction of the observation hole was completed, we sent the $SF_6$ transmitter into the hole and used the Φ10 mm PVC pipe and the seamless steel pipe to seal the hole. The sealing form was "two plugging and one injection," and the sealing depth was 20 m. After the sealing work was completed, we installed the pressure gauge and gas concentration recorder. After the liquid $CO_2$ phase change blasting was completed, we used the same method to seal the hole, and the end of the steel pipe was kept breathable. After 24 hours, we injected the tracer gas SF6 into the cracking hole and began the record.

## 4.2 Analysis of effect

During the observation period, the gas outflow and inflow in the observation hole were dynamic. Therefore, according to this comprehensive study, we used the gas inflow stage—in particular, the previous data—as the main calculation parameter in the process of gas inflow velocity. Through data processing, we obtained the changing process of $SF_6$ gas flow parameters in the observation hole, as shown in Figs 10 and 11. The data of the permeability coefficient λ obtained from the 12 observation holes are given in Table 7.

From the data shown in Figs 10 and 11, it is evident that $SF_6$ gas gradually diffused to each observation hole after being injected into the cracking hole. Among them, the $SF_6$ gas signal

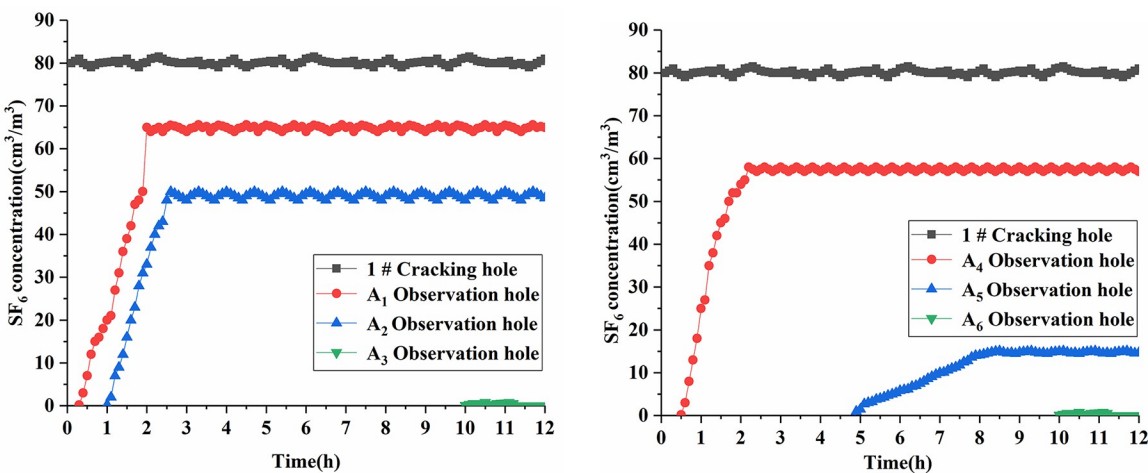

**Fig 10. $SF_6$ concentration curve of No. 1 test hole.**

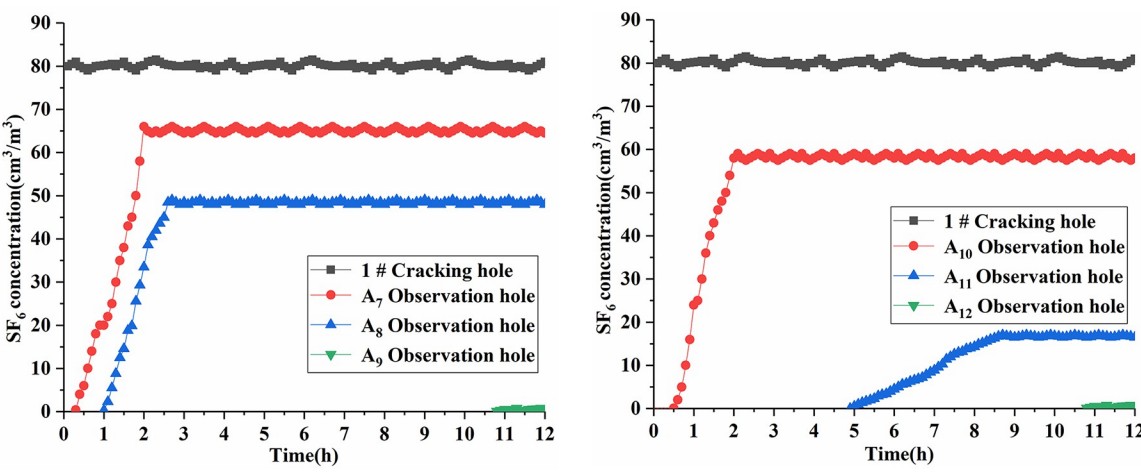

**Fig 11. $SF_6$ concentration curve of No. 2 test hole.**

was measured in a short time in the $A_1$ and $A_7$ observation holes 1 m away from the crack hole. The $A_1$ and $A_7$ observation holes were close to the broken zone of coal damage after liquid $CO_2$ phase change blasting, and the coal body formed a large crack under the action of detonation gas. As a result, the $SF_6$ gas could be transmitted quickly. The time of the $SF_6$ gas signal measured by the $A_4$ and $A_{10}$ observation holes 1.5 m away from the fracture hole was later than that of the observation hole 1 m away from the fracture hole. The stable $SF_6$ gas concentration was relatively reduced, mainly because the $A_1$ and $A_7$ observation holes were in the middle of the blasting area, and the influence of the blasting stress was relatively reduced. The fracture development effect in this area was better, which was helpful to the conduction of $SF_6$ gas. The $A_2$ and $A_8$ observation holes with a distance of 2 m from the cracking hole had a late time to measure the $SF_6$ gas signal, and the stable $SF_6$ gas was less concentrated. The $A_2$ and $A_8$ observation holes were located in the crack propagation area, which was the end area affected by the liquid $CO_2$ phase change blasting. This area was the blasting stress wave energy attenuation area, and the degree of coal body crack propagation was small. As a result, the $SF_6$ gas conduction speed was slow and the concentration was reduced. The $A_5$ and $A_{11}$ observation holes 2.5 m away from the cracking hole received a very low concentration of $SF_6$ gas signal after a long period of time, mainly because of the vibration zone after the liquid $CO_2$ phase change blasting at the $A_5$ and $A_{11}$ observation holes. The stress wave in this area was attenuated to such an extent that it no longer caused damage to the coal body. Because of the short distance from the blasting influence range, however, the $SF_6$ gas was transmitted along the coal body fissure to the observation hole. The $A_3$, $A_6$, $A_9$, and $A_{12}$ observation holes, which were more than 2.5 m away from the crack holes, appeared to have a weak $SF_6$ gas signal in the later period of measurement. These holes were far away from the effective influence range of the liquid $CO_2$ phase change blasting, and only a small amount of $SF_6$ gas conduction to the original crack of coal. The farther the observation hole is from the cracking hole, the slower the conduction velocity of $SF_6$ and the lower the concentration content. The stress wave had different effects on the development of the coal cracks in different regions. The crack conduction law of the liquid $CO_2$ phase change blasting in the field was consistent with the theoretical analysis results. The effective distribution range of the farthest tracer gas $SF_6$ was between 2 m and 2.5 m from the cracking hole, which proved that the liquid $CO_2$ phase change blasting crack radius of the two measuring points in the west return airway of the No. 5 mining area of the Sihe Coal Mine was between 2 m and 2.5 m. Therefore, the maximum error of the two test points

**Table 7. Coal seam permeability coefficient at different distances.**

| Distance (m) | | 1 | 1.5 | 2 | 2.5 | 3 | 3.5 |
|---|---|---|---|---|---|---|---|
| Coal seam permeability coefficient (m²/MPa²·d) | 1#Cracking hole | 2.97 | 1.55 | 0.23 | 0.0332 | 0.0332 | 0.0332 |
| | 2#Cracking hole | 2.57 | 1.45 | 0.18 | 0.0332 | 0.0332 | 0.0332 |

calculated by the calculation and the prediction model was 5.93% < 20%. This was considered to be a small error, which further proved that the prediction model was reasonable.

The permeability coefficient of the original coal seam in the west return airway of the No. 5 mining area of the Sihe Coal Mine was 0.0332 $m^2 \cdot (MPa^2 \cdot d)^{-1}$. From the data given in Table 7, it is evident that the permeability coefficient of coal seam after liquid $CO_2$ phase change cracking increased to 2.97 $m2 \cdot (MPa^2 \cdot d)^{-1}$. We took the data of each measuring point in the influence range of liquid $CO_2$ phase change blasting as the average value and determined that the permeability coefficient of the coal seam had increased by 45 times. Thus, the antireflection effect of liquid $CO_2$ phase change blasting in the west return airway of the No. 5 mining area of the Sihe Coal Mine was obvious.

## 5. Conclusion

On the basis of this study, we achieved the following results:

1. The stress wave propagation mechanism and crack propagation mechanism of liquid $CO_2$ phase change blasting were revealed. The blasting process was divided into crushing zones under the action of strong impact energy in the initial stage. We obtained a blasting middle zone of crack initiation that was caused by stress wave attenuation; a crack propagation zone in which the crack continued to expand in the later stage of blasting; and a vibration zone that transmitted in the form of vibration waves without causing damage to the coal body. Through TNT equivalent conversion, the energy release equivalent of liquid $CO_2$ phase change blasting cracker was about 588 g TNT.

2. Based on the orthogonal design of the four factors affecting the cracking effect, we conducted a numerical simulation of liquid $CO_2$ phase change cracking using ANSYS-LSDYNA numerical simulation software. Through range analysis and variance analysis, we obtained the primary and secondary factors affecting the cracking radius of liquid $CO_2$ phase change blasting. The order was ground stress, gas pressure, coal body firmness coefficient, and gas content. The cracking radius decreased with the increase of ground stress and increased with an increase in gas pressure, coal body firmness coefficient, and gas content.

3. Based on the multiple regression analysis of the numerical simulation results, we established a prediction model of the fracture radius of liquid $CO_2$ phase change blasting under the coupling conditions of four different factors (i.e., ground stress, gas pressure, coal firmness coefficient, and gas content). We conducted a double verification of two sets of numerical simulation and field industrial test. The measured crack radius range of liquid $CO_2$ phase change blasting was 2 m to 2.5 m, of which the maximum error of numerical simulation was 2.8%, and the maximum error of field industrial test was 5.93%. The error was small, which proved that the prediction model of predicting the crack radius of liquid $CO_2$ phase change blasting was correct. This prediction model could be used to predict the field blasting range, which had a certain reference value for the study of gas antireflection hole distribution.

4. We conducted the $SF_6$ tracer gas test in liquid $CO_2$ phase change blasting field. The results showed that the farther away it was from the cracking hole, the slower the $SF_6$ conduction velocity and the lower the concentration. Because of the attenuation of stress wave after blasting, different effects of crack development occurred in different regions. The crack

propagation law of liquid $CO_2$ phase change blasting was consistent with the theoretical analysis results. The permeability coefficient of coal seam increased by 45 times after liquid $CO_2$ phase change blasting. The effect of liquid $CO_2$ phase change blasting in the west return airway of the Sihe Coal Mine was obvious.

## Supporting information

**S1 Table. Formula symbol summary table.**
(DOCX)

## Author Contributions

**Conceptualization:** Jinzhang Jia, Yumo Wu.

**Data curation:** Jinzhang Jia, Yumo Wu, Bin Li.

**Formal analysis:** Yumo Wu, Bin Li.

**Funding acquisition:** Jinzhang Jia.

**Investigation:** Jinzhang Jia, Yumo Wu, Dan Zhao.

**Methodology:** Jinzhang Jia, Yumo Wu.

**Software:** Yumo Wu, Dan Zhao, Bin Li.

**Validation:** Jinzhang Jia, Dan Zhao, Dongming Wang.

**Visualization:** Yumo Wu, Bin Li.

**Writing – original draft:** Jinzhang Jia, Yumo Wu, Dan Zhao, Bin Li.

**Writing – review & editing:** Yumo Wu, Dan Zhao.

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
