## [Decision Letter · Decision Letter 0]

2 Dec 2022

PONE-D-22-27273Study on Prediction of Blasting Cracking Radius of Liquid CO2 in CoalPLOS ONE

Dear Dr. Wu,

Thank you for submitting your manuscript to PLOS ONE. After careful consideration, we feel that it has merit but does not fully meet PLOS ONE’s publication criteria as it currently stands. Therefore, we invite you to submit a revised version of the manuscript that addresses the points raised during the review process.

We look forward to receiving your revised manuscript.

Kind regards,

Yanping Yuan

Academic Editor

PLOS ONE

Journal Requirements:

2. Please note that PLOS ONE has specific guidelines on code sharing for submissions in which author-generated code underpins the findings in the manuscript. In these cases, all author-generated code must be made available without restrictions upon publication of the work. Please review our guidelines at https://journals.plos.org/plosone/s/materials-and-software-sharing#loc-sharing-code and ensure that your code is shared in a way that follows best practice and facilitates reproducibility and reuse. New software must comply with the Open Source Definition.

"This work was partly supported by the National Natural Science Foundation of China (grant number 52174183), the Natural Science Foundation of Liaoning Province (grant number 2019-MS-162) and the Scientific Research Project of Guangdong Provincial Department of Education—Young Innovative Talents Project（grant number 2022KQNCX141）."

 "This work was partly supported by the National Natural Science Foundation of China (grant number 52174183), the Natural Science Foundation of Liaoning Province (grant number 2019-MS-162) and the Scientific Research Project of Guangdong Provincial Department of Education—Young Innovative Talents Project（grant number 2022KQNCX141）. The first and second funders are first author Jinzhang Jia and the third funder is third author Dan Zhao."

"The authors declare that they have no known competing financial interests or personal relationships that could have appeared to influence the work reported in this paper."

Reviewers' comments:

Reviewer's Responses to Questions

**Comments to the Author**

1. Is the manuscript technically sound, and do the data support the conclusions?

Reviewer #1: Yes

Reviewer #2: No

2. Has the statistical analysis been performed appropriately and rigorously? 

Reviewer #1: Yes

Reviewer #2: No

3. Have the authors made all data underlying the findings in their manuscript fully available?

Reviewer #1: Yes

Reviewer #2: No

4. Is the manuscript presented in an intelligible fashion and written in standard English?

Reviewer #1: Yes

Reviewer #2: No

5. Review Comments to the Author

Reviewer #1: Based on the ANSYS / LS-DYNA numerical simulation method, the author reveals the influence mechanism of different factors on the fracture radius of liquid CO2 phase change blasting, and establishes a prediction model for predicting the fracture radius of liquid CO2 phase change blasting, which has achieved good results.The conclusion has reflected their analysis and the manuscript is in a proper format to understand the content very well.Recommended for publication in journals after minor repairs.

Questions to Authors:

1.Eliminate multiple references. After thatplease check the manuscript thoroughly and eliminate all the lumps in the manuscript.This should be done by characterising eachreference individually This can be done by mentioning 1or 2 phrases per reference to show how it is different from the others andwhy it deserves mentioning.

2.In Section 2.1, there is a reference number sorting problem, and it is recommended to correct this error.

3.Eliminate the use of redundant words e.g. in this way, recently,respectively, therefore,currently, thus, hence, finally, to do this, first,in order, however, moreover,nowadays consequently, in addition,additionally, furthermore. Revise all similar cases, asremoving these term(s)would not significantly affect the meaning of the sentence.

4.The deionof"Introduction"still needs to beimproved, as some contents are redundancy while some are lacked.

5.Consult the journal's reference style for theexact appearance of these elements, anduse of punctuation and capitalization. Cite the articles published in recent years, just as below:

[1] Liu SM, Li XL, Wang DK. et al. Investigations on the mechanism of the microstructural evolution of different coal ranks under liquid nitrogen cold soaking, Energy Sources, Part A: Recovery, Utilization, and Environmental Effects. 2020, 1-17. https://doi.org/10.1080/15567036.2020.1841856

[2] Zhou XM, Wang S, Li XL. Research on theory and technology of floor heave control in semicoal rock roadway: Taking longhu coal mine in Qitaihe mining area as an Example. Lithosphere. 2022, 2022(11): 3810988. https://doi.org/10.2113/2022/3810988

[3] Wang, S. Li XL, Qin QZ. Study on surrounding rock control and support stability of Ultra-large height mining face. Energies. 2022, 15(18): 6811. https://doi.org/10.3390/en15186811

[4] Li XL, Chen SJ, Wang S. Study on in situ stress distribution law of the deep mine taking Linyi Mining area as an example, Advances in Materials Science and Engineering, 2021, 9(4): 5594181. https://doi.org/10.1155/2021/5594181

[5] Liu HY, Zhang BY, Li XL. Research on roof damage mechanism and control technology of gob-side entry retaining under close distance gob, Engineering Failure Analysis, 2022, 138(5), 106331. https://doi.org/10.1016/j.engfailanal.2022.106331

Reviewer #2: On the basis of careful reading of the literature, this paper studies the prediction of the blasting radius of liquid Co2 in coal. I think the innovation of this paper is general. I have the following opinions :

1.Carbon dioxide phase change rock breaking is not a very new research direction. Many scholars have done a lot of work. The author needs to clarify the innovation of the paper and distinguish it from the latest research.

2. About the phase transition mechanism of carbon dioxide used in this paper, the author lacks the basic knowledge of supercritical phase transition, and the basic parameters and boundary conditions of the simulation model are not clear enough.

3. Although the paper carried out experimental and simulation analysis, but the experimental phenomena and phase transition mechanism of the discussion is too little, the paper only contains the presentation of data results.

4. The numerical simulation picture is not clear enough, some values are covered by the picture, please use the original picture.

5. The use of curves is not standardized.

6. For the test and simulation results, the author needs to summarize a function or equation.

7.There are many symbols used in the formula, it is recommended to make a table to summarize.

Overall, I think this paper needs to be carefully modified to consider whether to meet the standards of the journal.

6. PLOS authors have the option to publish the peer review history of their article (what does this mean?). If published, this will include your full peer review and any attached files.

Reviewer #1: No

Reviewer #2: No

---

## [Author Response · Author response to Decision Letter 0]

26 Dec 2022

Replies to the reviewers’ comments: 

Reviewer #1:

1. Eliminate multiple references. After thatplease check the manuscript thoroughly and eliminate all the lumps in the manuscript.This should be done by characterising eachreference individually This can be done by mentioning 1or 2 phrases per reference to show how it is different from the others andwhy it deserves mentioning. 

Response: Eliminates multiple references; No more than 3 references per citation. Modify as follows :

In this paper, the liquid CO2 phase change blasting antireflection technology is described. This method has the advantages of low investment and high efficiency. When it is gradually popularized and applied in the field of coalbed methane development, many references are cited. After careful study, two redundant references were deleted, which were Literature 17 and Literature 18.

[17] Karmis M, Ripepi N, Gilliland E, et al. Central appalachian basin unconventional (coal/organic shale) reservoir small scale CO2 injection test. United States: N. p.: Web; 2018. doi:10.2172/1439921.

[18] Pan,Ye,Zhou,Tan,Connell,Fan. CO2 storage in coal to enhance coalbed methane recovery: a review of field experiments in China[J]. International Geology Review,2018,60(5-6). doi:10.1080/00206814.2017.1373607.

2. In Section 2.1, there is a reference number sorting problem, and it is recommended to correct this error. 

Response: Thank the reviewer for raising this question. In Section 2.1, I found a redundant reference that I deleted from the manuscript and reordered the remaining references.

3. Eliminate the use of redundant words e.g. in this way, recently, respectively, therefore, currently, thus, hence, finally, to do this, first, in order, however, moreover, nowadays consequently, in addition, additionally, furthermore. Revise all similar cases, asremoving these term(s)would not significantly affect the meaning of the sentence. 

Response: Thanks for reading my manuscript carefully by the reviewer, I have deleted some redundant words.

4. The deionof"Introduction"still needs to beimproved, as some contents are redundancy while some are lacked. 

Response: According to the requirements of the reviewer, I modified the introduction and deleted some redundant references.

5. Consult the journal's reference style for theexact appearance of these elements, anduse of punctuation and capitalization. Cite the articles published in recent years, just as below: 

[1] Liu SM, Li XL, Wang DK. et al. Investigations on the mechanism of the microstructural evolution of different coal ranks under liquid nitrogen cold soaking, Energy Sources, Part A: Recovery, Utilization, and Environmental Effects. 2020, 1-17. https://doi.org/10.1080/15567036.2020.1841856

[2] Zhou XM, Wang S, Li XL. Research on theory and technology of floor heave control in semicoal rock roadway: Taking longhu coal mine in Qitaihe mining area as an Example. Lithosphere. 2022, 2022(11): 3810988. https://doi.org/10.2113/2022/3810988

[3] Wang, S. Li XL, Qin QZ. Study on surrounding rock control and support stability of Ultra-large height mining face. Energies. 2022, 15(18): 6811. https://doi.org/10.3390/en15186811

[4] Li XL, Chen SJ, Wang S. Study on in situ stress distribution law of the deep mine taking Linyi Mining area as an example, Advances in Materials Science and Engineering, 2021, 9(4): 5594181. https://doi.org/10.1155/2021/5594181

[5] Liu HY, Zhang BY, Li XL. Research on roof damage mechanism and control technology of gob-side entry retaining under close distance gob, Engineering Failure Analysis, 2022, 138(5), 106331. https://doi.org/10.1016/j.engfailanal.2022.106331. 

Response: Thank the reviewer for reading my manuscript carefully. I have modified the format of the reference according to the requirements of the journal and cited several articles published in recent years, including:

[6] Wang, S. Li XL, Qin QZ. Study on surrounding rock control and support stability of Ultra-large height mining face. Energies. 2022, 15(18): 6811. https://doi.org/10.3390/en15186811

[18] Liu SM, Li XL, Wang DK. et al. Investigations on the mechanism of the microstructural evolution of different coal ranks under liquid nitrogen cold soaking, Energy Sources, Part A: Recovery, Utilization, and Environmental Effects. 2020, 1-17. https://doi.org/10.1080/15567036.2020.1841856

[33] Zhou XM, Wang S, Li XL. Research on theory and technology of floor heave control in semicoal rock roadway: Taking longhu coal mine in Qitaihe mining area as an Example. Lithosphere. 2022, 2022(11): 3810988. https://doi.org/10.2113/2022/3810988

[34] Li XL, Chen SJ, Wang S. Study on in situ stress distribution law of the deep mine taking Linyi Mining area as an example, Advances in Materials Science and Engineering, 2021, 9(4): 5594181. https://doi.org/10.1155/2021/5594181

[35] Liu HY, Zhang BY, Li XL. Research on roof damage mechanism and control technology of gob-side entry retaining under close distance gob, Engineering Failure Analysis, 2022, 138(5), 106331. https://doi.org/10.1016/j.engfailanal.2022.106331. 

Reviewer #2:

1. Carbon dioxide phase change rock breaking is not a very new research direction. Many scholars have done a lot of work. The author needs to clarify the innovation of the paper and distinguish it from the latest research..

Response: At present, many scholars have carried out some research on liquid CO2-phase transition fracturing technology. Many factors influence the liquid CO2 phase transition fracturing radius due to the complex geographical environment of the underground. If multiple factors are considered, the calculation amount will be huge. Most of the existing research is limited to the influence of a single factor on the fracturing radius, and the calculation methods are different, so the accuracy of predicting the liquid CO2 phase transition fracturing radius is not high. The innovation of this paper is to use the orthogonal design method to reduce the simulation workload and establish a prediction model of liquid CO2 phase change blasting radius affected by various factors. This research method can provide a theoretical reference for predicting the range of fracturing radii of different blasting methods. I described the specific modifications in the introduction part of the text as follows:

Many scholars have conducted some research on liquid CO2 phase change cracking technology, but most of the existing research has been limited to the influence of single factor on the cracking radius, and the calculation methods are different. Because of the complex geographical environment of the underground, many factors affect the cracking radius of liquid CO2 phase change, and the propagation law of cracks after blasting is relatively complex. This requires the application of reasonable methods to conduct multifactor coupling analysis, and the calculation method of liquid CO2 phase change cracking radius is optimized to enhance the prediction accuracy and improve the controllability of crack propagation after blasting.

In light of this research, under the condition of considering the coupling of four groups of influencing factors, such as ground stress, gas pressure, coal body firmness coefficient, and gas content, we used ANSYS-LSDYNA numerical simulation software to analyze the primary and secondary order and influence degree of the factors affecting the cracking radius of liquid CO2 phase change blasting by combining range and variance analysis. We used an orthogonal design method to optimize the numerical simulation scheme and reduce the simulation workload. We processed the simulation data using multiple regression analysis and established a prediction model of the cracking radius of liquid CO2 phase change blasting under the influence of various factors. This research method provides a theoretical reference for the prediction of cracking radius of different blasting methods.

2. About the phase transition mechanism of carbon dioxide used in this paper, the author lacks the basic knowledge of supercritical phase transition, and the basic parameters and boundary conditions of the simulation model are not clear enough. 

Response: The basic knowledge of supercritical phase transition is added to the manuscript for the carbon dioxide phase transition mechanism used in this paper. The specific modifications are as follows:

The characteristics of CO2 at ambient temperature and pressure are easy to liquefy and not to burn. The critical condition of CO2 is 7.38 MPa and the temperature is 31.1°C. If the temperature exceeds the critical temperature, the liquid CO2 absorbs heat and enters the supercritical state. Supercritical fluid has the advantages of a gas-liquid two-phase fluid at the same time. It has super flow, transmission, and permeability. It can maximize the conductivity of natural fractures and improve the conductivity of fractures. It has unique advantages in fracturing. The basic principle of liquid CO2 cracking technology is as follows: liquid CO2 is loaded into the cracking device and placed in the drilled hole, and the equipped cracking device is connected with the mine initiator to form a closed loop. After detonation, the hot cartridge will release heat energy instantaneously under the excitation of a current. This increase in heat makes the liquid CO2 enter the supercritical state immediately. In this state, the rapid gasification volume of liquid CO2 increases by more than 700 times in an instant, and the pressure in the cracking device increases rapidly. When the gas pressure in the cracking device reaches a predetermined value, the preset shear disc at the venting head breaks, and the gaseous CO2 erupts into the coal around the cracking device through the guide hole. This instantly produces strong impact pressure.

The basic parameters and boundary conditions of the simulation model are supplemented and modified. The specific modifications are as follows:

The model consists of three parts: explosive, air, and coal rock. The fluid-solid coupling algorithm is adopted. The explosive and air use the Euler algorithm, the coal rock uses the Lagrange algorithm, the unit uses the multi-material ALE algorithm, and the continuous medium model uses the ALG algorithm. The mesh of single-hole blasting is divided into 369411 units, and the number of nodes is 742344. The mesh of double-hole blasting is divided into 859197 units, and the number of nodes is 1724102. A uniform load of 12 MPa is applied to the top of the model, which is equivalent to the weight of a 900-meter rock layer. The lateral pressure coefficient is 1.5; that is, the horizontal stress is 18 MPa. The boundary conditions are: the surface of the coal-rock model is constrained, the sides of model Z are unidirectionally constrained, the Z direction is constrained, and the model boundary is a non-reflective interface boundary.

3. Although the paper carried out experimental and simulation analysis, but the experimental phenomena and phase transition mechanism of the discussion is too little, the paper only contains the presentation of data results.

Response: Thank you to the reviewers for pointing out the shortcomings of my paper. I also found this problem through the re-examination of the manuscript. There is too little discussion on the experimental phenomena and phase transition mechanism of the manuscript. I added the description and mechanism analysis of the simulation evolution process of liquid CO2 phase transition blasting. The specific modifications are as follows:

Because the stress and strain on coal under blasting is very complex, in order to show the propagation and distribution of the explosive stress field in the process of liquid CO2 phase change blasting, the damage cloud map of the model at different times can be dynamically described by LS-PREPOST post-processing software. Through the concealment of explosives and air units, the effective stress of coal material can be understood more intuitively. Through the evolution simulation of the whole process of blasting, the coal damage distribution cloud map of the blasting effect at different times is obtained, as shown in Fig. 4.

（a）t=20us （b）t=35us

（a）t=80us （b）t=100us

Fig. 4 Simulation evolution process of liquid CO2 phase transition blasting

The fracture development law of liquid CO2 phase change blasting in coal bodies is as follows: A strong stress wave will be generated in the borehole immediately after detonation in t = 0~20 us. Because the coal wall near the blasting hole absorbs a large amount of stress wave energy, the coal wall around the borehole is crushed over a large area; that is, a crushing area is formed. The main crack appears gradually when t = 20 us. When t = 35 us, branched cracks begin to appear, the area of the high pressure zone in the center of the borehole gradually increases, and new cracks gradually form. The stress wave continues to extend outward from the coal-rock crushing zone of the coal seam and gradually forms a coal-rock crack zone. When t = 20~80 us, the crack continues to expand. When t = 80 us, the damage zone of the fracture ring reaches its maximum, and the stress wave generated by the gas explosion has changed from a plastic to an elastic wave. The driving force of the crack to the distant extension is becoming smaller and smaller as high gas pressure is continuously attenuated. When t = 100us, the maximum axial stress at the tip of the extended crack is less than its own dynamic ultimate tensile strength, and the crack stops expanding. It can be seen that blasting gas damage to coal rock is a complex, dynamic evolution process.

4. The numerical simulation picture is not clear enough, some values are covered by the picture, please use the original picture.

Response: Thank you to the reviewers who read my manuscript carefully and found this subtle problem. Because I gave too much consideration to the aesthetics of layout when writing the manuscript, I arranged the simulated original pictures too tightly, resulting in the problem of occlusion of individual pictures. I have modified and re-uploaded these pictures.

5. The use of curves is not standardized.

Response: Because the reviewer pointed out the use of the curve, I corrected and modified all of the curves throughout the manuscript. The visual analysis diagram of the influence of various factors on the fracture radius in the curve of 6 contains four groups of factors: in-situ stress, gas pressure, coal firmness coefficient, and gas content. The units of the four groups of factors on the x-axis are different, but the range of the fracture radius on the y-axis is roughly the same. In order to observe the slope, numerical laws, and other laws, the primary and secondary orders and laws of the factors affecting the fracture radius of liquid CO2 phase change blasting are better compared and analyzed. Therefore, the origin software is used to combine the four groups of curves to better reach the effect of visual analysis.

6. For the test and simulation results, the author needs to summarize a function or equation.

Response: For the test and simulation results, it can be seen that the influence factors and simulation results can be described by a linear relationship. Therefore, the ground stress is x1, the gas pressure is x2, the coal firmness coefficient is x3, the gas content is x4, and the fracture radius is y. Thus, the prediction model equation between y and x is established：

7. There are many symbols used in the formula, it is recommended to make a table to summarize.

Response: At the request of the reviewer, I created a table that summarized all the symbols in the formula as follows:

Symbols Implication Unit

ρ density of coal kg·m-3

u、υ、w three displacement components of coal mass point m

 , , and 

hree acceleration components of particle m·s−2

σx, σy, σz, τxy, τxz, and τyz six stress components of coal (rock) mass point 

Eg gas explosion energy kJ

P1 gas pressure in the blasting cracker 275 MPa

P2 standard atmospheric pressure 0.10108 MPa

V cracking volume m3

K adiabatic index of the medium 

Q explosion energy of 1 kg TNT explosive 4250 kJ·kg−1

x1 ground stress MPa

x2 gas pressure MPa

x3, coal firmness coefficient 

x4 gas content m3·t-1

y liquid CO2 phase change blasting cracking radius m

---

## [Decision Letter · Decision Letter 1]

28 Dec 2022

PONE-D-22-27273R1Study on Prediction of Blasting Cracking Radius of Liquid CO2 in CoalPLOS ONE

Dear Dr. Wu,

Thank you for submitting your manuscript to PLOS ONE. After careful consideration, we feel that it has merit but does not fully meet PLOS ONE’s publication criteria as it currently stands. Therefore, we invite you to submit a revised version of the manuscript that addresses the points raised during the review process.

We look forward to receiving your revised manuscript.

Kind regards,

Yanping Yuan

Academic Editor

PLOS ONE

Journal Requirements:

Reviewers' comments:

Reviewer's Responses to Questions

**Comments to the Author**

1. If the authors have adequately addressed your comments raised in a previous round of review and you feel that this manuscript is now acceptable for publication, you may indicate that here to bypass the “Comments to the Author” section, enter your conflict of interest statement in the “Confidential to Editor” section, and submit your "Accept" recommendation.

Reviewer #1: All comments have been addressed

Reviewer #2: All comments have been addressed

2. Is the manuscript technically sound, and do the data support the conclusions?

Reviewer #1: Yes

Reviewer #2: Yes

3. Has the statistical analysis been performed appropriately and rigorously? 

Reviewer #1: Yes

Reviewer #2: N/A

4. Have the authors made all data underlying the findings in their manuscript fully available?

Reviewer #1: Yes

Reviewer #2: Yes

5. Is the manuscript presented in an intelligible fashion and written in standard English?

Reviewer #1: Yes

Reviewer #2: Yes

6. Review Comments to the Author

Reviewer #1: Dear authors. The manuscript has been changed according to my comments, it could be accepted in the state.

Reviewer #2: I hope that the author will make a summary table of all the symbols of the formula, and optimize the curve in the figure, so as to achieve the level of publication.

7. PLOS authors have the option to publish the peer review history of their article (what does this mean?). If published, this will include your full peer review and any attached files.

Reviewer #1: No

Reviewer #2: No

---

## [Author Response · Author response to Decision Letter 1]

30 Dec 2022

Dear editor: 

Thank you very much for giving us an opportunity to revise our manuscript. We appreciate the editor and reviewers very much for their constructive comments and suggestions on our manuscript entitled“Study on Prediction of Blasting Cracking Radius of Liquid CO2 in Coal”(ID: PONE-D-22-27273).

We have studied reviewers' comments carefully. According to the reviewers' detailed suggestions. we have made a careful revision on the original manuscript. I modified the copy of the markup to form a separate file, named 'Revised Manuscript with Track Changes'.

Kind regards.

Corresponding author: WU Yumo

E-mail address: 13614067811@163.com

Replies to the reviewers’ comments: 

Reviewer #2: I hope that the author will make a summary table of all the symbols of the formula, and optimize the curve in the figure, so as to achieve the level of publication.

Response: Because the reviewer pointed out the use of the curve, I corrected and modified all of the curves throughout the manuscript. The visual analysis diagram of the influence of various factors on the fracture radius in the curve of 6 contains four groups of factors: in-situ stress, gas pressure, coal firmness coefficient, and gas content. The units of the four groups of factors on the x-axis are different, but the range of the fracture radius on the y-axis is roughly the same. In order to observe the slope, numerical laws, and other laws, the primary and secondary orders and laws of the factors affecting the fracture radius of liquid CO2 phase change blasting are better compared and analyzed. Therefore, the origin software is used to combine the four groups of curves to better reach the effect of visual analysis. According to the standard general drawing format, the following is modified :

Fig. 6 Intuitive analysis of influence of factors on cracking radius

At the request of the reviewer, I created a table that summarized all the symbols in the formula as follows:

Table. 8 Formula symbol summary table

---

## [Decision Letter · Decision Letter 2]

8 Jan 2023

Study on Prediction of Blasting Cracking Radius of Liquid CO2 in Coal

PONE-D-22-27273R2

Dear Dr. Wu,

We’re pleased to inform you that your manuscript has been judged scientifically suitable for publication and will be formally accepted for publication once it meets all outstanding technical requirements.

Kind regards,

Yanping Yuan

Academic Editor

PLOS ONE

Additional Editor Comments (optional):

Reviewers' comments:

Reviewer's Responses to Questions

**Comments to the Author**

1. If the authors have adequately addressed your comments raised in a previous round of review and you feel that this manuscript is now acceptable for publication, you may indicate that here to bypass the “Comments to the Author” section, enter your conflict of interest statement in the “Confidential to Editor” section, and submit your "Accept" recommendation.

Reviewer #1: All comments have been addressed

Reviewer #2: All comments have been addressed

2. Is the manuscript technically sound, and do the data support the conclusions?

Reviewer #1: Yes

Reviewer #2: Yes

3. Has the statistical analysis been performed appropriately and rigorously? 

Reviewer #1: Yes

Reviewer #2: Yes

4. Have the authors made all data underlying the findings in their manuscript fully available?

Reviewer #1: Yes

Reviewer #2: Yes

5. Is the manuscript presented in an intelligible fashion and written in standard English?

Reviewer #1: Yes

Reviewer #2: Yes

6. Review Comments to the Author

Reviewer #1: The manuscript has been changed according to my comments, so I think it could be accepted in the state.

Reviewer #2: I think all revisions have been done. I think the manuscript is suitable to be considered for publication

7. PLOS authors have the option to publish the peer review history of their article (what does this mean?). If published, this will include your full peer review and any attached files.

Reviewer #1: No

Reviewer #2: No

---

## [Editor Report · Acceptance letter]

11 Jan 2023

PONE-D-22-27273R2 

Study on Prediction of Blasting Cracking Radius of Liquid CO2 in Coal 

Dear Dr. Wu:

I'm pleased to inform you that your manuscript has been deemed suitable for publication in PLOS ONE. Congratulations! Your manuscript is now with our production department. 

Kind regards, 

on behalf of

Prof. Yanping Yuan 

Academic Editor

PLOS ONE